# STOP Strategy to Inhibit *P. falciparum* and *S. aureus* Growth: Molecular Mechanism Studies on Purposely Designed Hybrids

**DOI:** 10.3390/antibiotics14100991

**Published:** 2025-10-03

**Authors:** Beatrice Gianibbi, Riccardo Corina, Nicoletta Basilico, Ottavia Spiga, Silvia Gobbi, Federica Belluti, Giovanna Angela Gentilomi, Silvia Parapini, Francesca Bonvicini, Alessandra Bisi

**Affiliations:** 1Department of Biotechnology, Chemistry and Pharmacy, University of Siena, Via Aldo Moro 2, 53100 Siena, Italy; b.gianibbi@gmail.com (B.G.); ottavia.spiga@unisi.it (O.S.); 2Department of Pharmacy and Biotechnology, Alma Mater Studiorum-University of Bologna, Via Belmeloro 6, 40126 Bologna, Italy; riccardo.corina@cirad.fr (R.C.); silvia.gobbi@unibo.it (S.G.); federica.belluti@unibo.it (F.B.); 3Department of Biomedical, Surgical and Dental Sciences, Università degli Studi di Milano, Via Carlo Pascal, 36, 20133 Milano, Italy; nicoletta.basilico@unimi.it; 4Department of Pharmacy and Biotechnology, Alma Mater Studiorum-University of Bologna, Via Massarenti 9, 40138 Bologna, Italy; giovanna.gentilomi@unibo.it; 5Microbiology Unit, IRCCS Azienda Ospedaliero-Universitaria di Bologna, Via Massarenti 9, 40138 Bologna, Italy; 6Department of Biomedical Sciences for Health, Università degli Studi di Milano, Via Carlo Pascal, 36, 20133 Milano, Italy; silvia.parapini@unimi.it

**Keywords:** malaria, bacterial infections, phenothiazine, chloroquine, molecular docking, quantum-mechanics

## Abstract

**Background/Objectives**: Malaria remains the most critical parasitic disease globally, responsible for over 600.000 deaths annually. In sub-Saharan Africa, co-infections of *Plasmodium falciparum* with other pathogens, particularly *Staphylococcus aureus*, are common in children with severe malaria. Therefore, the design of new compounds targeting both pathogens appears to be an urgent priority. **Methods**: A small series of hybrid compounds was designed and synthesized by linking the pharmacophore of the antimalarial drug chloroquine with the phenothiazine core. These compounds were tested in vitro against a panel of microbial strains and further analyzed through in silico simulations to predict their physical-chemical properties. **Results**: Compounds **4b** and **5b** emerged the most potent candidates of the series, showing a sub-micromolar inhibitory activity on *P. falciparum*, and a promising micromolar potency on *S. aureus* alongside with a low toxicity on mammalian cells. Molecular docking followed by molecular dynamics (MD) simulations identified the respiratory membrane NDH-2 enzyme as common target in both pathogens. **Conclusions**: Both experimental and computational findings provide compelling evidence for the use of the designed compounds in a STOP strategy, i.e., Same-Target-Other-Pathogen, to treat malaria and bacterial infections concurrently.

## 1. Introduction

The welfare of the global population is strongly affected by infectious diseases, and current treatments suffer deeply from the increasing resistance to existing drugs. This alarming trend is not abating, and, at present, only a few examples of new small molecules are entering the development pipeline to ensure the availability of effective drugs needed in the near future [1]. It is increasingly evident that new compounds should be designed to counteract the occurrence of resistance, engaging innovative potential targets and exploiting underexplored mechanisms of action [2,3].

Malaria, the most important parasitic disease worldwide, is a vector-borne communicable disease caused by different protozoan *Plasmodium* species and transmitted by the bite of an infected female Anopheles mosquito. Of the different species of *Plasmodium* that infect humans, *Plasmodium falciparum* is responsible for the majority of severe clinical malaria and deaths. According to the 2023 World Malaria Report, malaria caused 249 million cases and 608.000 deaths in 2022 (World Malaria Report 2023). An estimated 95% of all malaria deaths occur in the WHO African Region, where the transmission intensity of the disease is extremely high. Children are one of the most vulnerable groups for malaria morbidity and mortality, considering that 76% of deaths occur in children under 5 years of age (WHO, Malaria 2024). Moreover, a significant collection of data suggests that patients with *P. falciparum* malaria are more likely to develop concomitant invasive bacterial infections, even if the cause-and-effect relationship still remains unclear. The accumulated evidence suggests that the prevalence of Gram-positive organisms exceeded that of Gram-negative organisms, and *Staphylococcus aureus* emerged as the main cause of bacteremia, increasing the risk of mortality [4,5]. A possible strategy to counteract this complex drawback could be the design of a single molecule that simultaneously affects both pathogens. From a medicinal chemistry perspective, this goal could be pursued by taking advantage of the so-called ‘covalent biotherapy’, which involves linking the scaffolds of well-known drugs that act through different mechanisms. This strategy could also lead to overcoming potential resistance [6]. An alternative approach could be the Same-Target-Other-Pathogen (STOP) strategy [2], which involves targeting homologous enzymes from different pathogens and has proven to be the best approach when the enzyme lacks a corresponding human homologue, thus ensuring high selectivity.

The quinoline moiety is a key structural constituent of many natural products and can be regarded as a privileged structure in medicinal chemistry: Chemical modifications performed on this core structure have led to derivatives endowed with different promising biological effects, among which antiprotozoal activity plays a central role [7]. Indeed, since the discovery of chloroquine (CQ) and its congeners and of the bis-piperazinyl-quinoline piperaquine (PPQ) (Figure 1), the quinoline core has attracted scientists’ attention and has been regarded as an ideal scaffold to be properly modified due to its commercial and synthetic availability, relatively safe profile, and low cost [8].

Moreover, the CQ mechanism of action against *P. falciparum*, albeit not completely clarified at the molecular level, seems to involve the quinoline moiety, and a vast number of papers dealing with suitably modified quinoline derivatives have been reported in the literature in recent years, aiming at finding new antimalarials able to evade CQ resistance [9,10]. In addition to the well-established significance of the quinoline scaffold in the development of antimalarial agents, its versatility has gained increasing attention in recent years. In particular, taking advantage of its chemical accessibility [11], the quinoline core has also been explored for its antibacterial potential. Notably, several quinoline-based analogues have demonstrated appreciable activity against a range of pathogens, including methicillin-resistant *S. aureus* (MRSA) [12,13]. The precise mode of action is still unknown, although the cell membrane appears to be the most likely target, as it encompasses crucial enzymes involved in essential metabolic functions [14].

The phenothiazine (PTZ) core, typically found in antipsychotic drugs, is now considered one of the most important privileged substructures in medicinal chemistry. Among the different biological activities reported, the antimicrobial potential of this tricyclic core has long been known [15]. In detail, a number of PTZ-based compounds have been reported as endowed with antifungal [16], antimicrobial [17], and antiprotozoal [18] activities, and due to their broad biological profile, this structure has been joined to different scaffolds, allowing the design of hybrid compounds [19,20,21]. Regarding its mechanism of action, the PTZ core has been recognized to act on microbial membranes, either by disrupting the pH gradient across membranes [22] or by inhibiting type II NADH dehydrogenase/NADH:quinone oxidoreductase (NDH-2), a key respiratory enzyme in many organisms. This enzymatic protein is tightly bound to the membrane by means of amphiphilic helices in the C-terminus and is responsible for the transfer of electrons from NADH to a quinone via FAD as a cofactor [23]. Through this mechanism, it helps maintain the NADH/NAD+ redox balance. The NDH-2 enzymatic complex is expressed in different microbial organisms, including *P. falciparum* and *S. aureus*, but not in mammals, making it an attractive target for selective antimicrobial drug development. Despite retaining the same enzymatic function, NDH-2 isoforms display different subcellular locations when comparing the mentioned pathogens: while the protein monotopically associates with the inward side of the plasmatic membrane in *S. aureus* [24], it conversely occupies the outward side of the inner mitochondrial membrane in *P. falciparum* [25]. Such diversity in the localization inevitably reflects a different accessibility of the enzymatic complex. A number of studies indicate that many chemical scaffolds as able to interact with NDH-2, and among them, quinolones and PTZs are the most commonly characterized enzyme inhibitors [26]. Recently, the crystal structure of NDH-2 complexed with the quinolone inhibitor 2-heptyl-4-hydroxyquinoline-N-oxide (HQNO) has been solved, providing new insight into the inhibition mechanism and identifying a previously unexplored hydrophobic cleft, which may help in the design of new inhibitors [27].

In this paper, a small series of hybrid compounds was designed and synthesized, in which the 7-chloro-4-amino-quinoline moiety found in CQ and PPQ was combined through a selected linker with a PTZ core, with the aim of obtaining compounds able to fight malaria co-infections. Indeed, this hybrid molecule could benefit from the multifunctional profiles of both scaffolds, thus exploiting the ‘covalent biotherapy’ strategy. Moreover, among the different bioactivities reported for the quinoline nucleus, the inhibition of the NDH-2 enzyme was also recently highlighted [2], setting the stage for an application of the STOP strategy. In detail, the PTZ nitrogen was linked to the alkylpiperazinyl–quinoline moiety of PPQ to give compounds **1**–**3** (Figure 2), and the copper-catalyzed azide-alkyne cycloaddition (CuAAC) was applied to insert a 1,2,3 triazole ring linking the chloroquinoline core of CQ (compounds **4a**–**6a** and **7**, Figure 2) or the alkylpiperazinyl–chloroquinoline fragment of PPQ (compound **8**, Figure 2) to the PTZ framework. Notably, the triazole moiety has recently been reported to be endowed with antiparasitic activity as well [28]. Preliminary results on both plasmodium and bacteria showed the best activity for the triazole-bearing compounds **4a**–**6a**. Considering that chlorpromazine, a well-known antipsychotic drug bearing a 2-chloroPTZ nucleus has been shown to potentiate CQ action against resistant strains of *P. falciparum* when concurrently administered [29], a chlorine atom was introduced in position 2 of the PTZ core in the most promising derivatives, to obtain **4b**–**6b**.

Compounds **1**–**3** were synthesized as reported in Figure 1. 10H-phenothiazine was alkylated with the selected dibromoalkane, and the resulting bromoalkyl intermediates **9**–**11** were reacted with 7-chloro-4-(piperazin-1-yl)quinoline [30] to obtain the desired compounds.

The 1,2,3 triazole-containing derivatives **4**–**5**(**a**,**b**) and **7** were prepared as outlined in Figure 2. 10-(prop-2-yn-1-yl)-10H-phenothiazine **12a** or 2-chloro-10-(prop-2-yn-1-yl)-10H-phenothiazine **12b**, prepared as previously reported by us [31], were submitted to CuAAC with N-(2-azidoethyl)-7-chloroquinolin-4-amine, N-(3-azidopropyl)-7-chloroquinolin-4-amine [32], or 4-azido-7-chloroquinoline [33]. For compound **8**, the click chemistry reaction was performed in the same conditions, starting from 10-(prop-2-yn-1-yl)-10H-phenothiazine **12a** and 4-(4-(3-azidopropyl)piperazin-1-yl)-7-chloroquinoline **14**, obtained from 7-chloro-4-(4-(3-chloropropyl)piperazin-1-yl)quinoline **13**, in turn prepared by alkylation of 7-chloro-4-(piperazin-1-yl)quinoline with 1-bromo-3-chloropropane, followed by reaction with NaN_3_.

For triazole derivatives **6a**,**b**, a different synthetic procedure was followed, as depicted in Figure 3. N-(5-bromopentyl)phthalimide was treated with NaN_3_ and the obtained 1-azido-5-phthalimidopentane 15 [34] underwent the CuAAC reaction with 10-(prop-2-yn-1-yl)-10H-phenothiazine or 2-chloro-10-(prop-2-yn-1-yl)-10H-phenothiazine to obtain the triazole phtalimide derivatives **16a**,**b**. Following the Ing-Manske hydrazinolysis, the N-substituted phthalimides were converted to the primary amines **17a**,**b**, which were then alkylated with 4,7 dichloroquinoline.

## 2. Results

All the final compounds **1**–**3**, **4**–**6a**,**b**, and **7**–**8** were screened for their in vitro antimicrobial activity against bacterial strains such as *Staphylococcus aureus* and *Escherichia coli*, and a fungal species, namely *Candida albicans,* whereas the antimalarial activity was assayed against *Plasmodium falciparum*. In addition, a comprehensive evaluation of the cytotoxicity of the PTZ-quinoline hybrid compounds was performed on human fibroblasts and human red blood cells.

### 2.1. In Vitro Antimicrobial Activity

The new PTZ-quinoline hybrid compounds **1**–**3**, **4**–**6a**,**b**, and **7**–**8** were assayed against *S. aureus*, *E. coli*, and *C. albicans* using the broth microdilution method by measuring the minimum inhibitory concentration (MIC). The commercial antimicrobial drugs gentamicin, ampicillin, and fluconazole served as standard controls in the experimental assays. MIC values are reported in Appendix A. While none of the synthesized derivatives proved effective against *E. coli* and *C. albicans*, even when tested at the highest concentration (100 µM), the triazole derivatives **4a**,**b**, **5a**,**b**, and **6b** displayed inhibitory activity against *S. aureus*, with **6b** being the least potent (MIC = 50 µM). Interestingly, among this set of compounds, all the chloro-substituted derivatives showed an increase in potency with respect to the unsubstituted derivatives, proving the significant contribution of this atom in the PTZ core for antibacterial activity; this effect became even more pronounced when the IC_50_ values for *S. aureus* were measured and plotted on an X/Y-axis graph (Table 1 and Appendix A); in particular, the chloro-substituted derivative **4b** displayed significantly higher inhibitory activity compared to **4a**, as demonstrated by the unpaired t-test, while the differences between **5a** and **5b** were not at a significant level (Appendix A).

Having demonstrated the potent anti-staphylococcal activity of compounds **4**–**5a**,**b**, five clinical isolates of methicillin-resistant *S. aureus* (MRSA) with different antibiotic resistance profiles were tested, and the results were compared with those obtained for the reference strain (Appendix A). It is worth noting that the IC_50_ values for these isolates were close to those of the *S. aureus* ATCC 25923, suggesting the effectiveness of the compounds not only against a laboratory strain but also against bacteria circulating in the human population.

### 2.2. In Vitro Antimalarial Activity

Compounds **1**–**3**, **4**–**6a**,**b**, and **7**–**8** were tested to evaluate their in vitro antiplasmodial activity against the *P. falciparum* CQ-sensitive (CQ-S) D10 and CQ-resistant (CQ-R) W2 strains, and compared to CQ as a standard drug. The results are collected in Table 1. With the exception of compounds **2** and **7**, all the PTZ-quinoline hybrid compounds exhibited antiplasmodial activity within the low micromolar or nanomolar range. Compounds **1**–**3** and **8**, incorporating the PTZ moiety linked to PPQ, demonstrated the lowest activity.

Compounds **4a**,**b**, **5a**,**b**, and **6a**,**b**, which contain the chloroquinoline core of CQ, exhibited activity within the nanomolar range. None of the tested compounds showed cross-resistance with CQ, demonstrating similar IC_50_ values against *P. falciparum* D10 and W2 strains. Interestingly, chloro-substituted derivatives exhibited enhanced activity, consistent with prior observations of antibacterial activity. While all compounds were less potent than CQ against CQ-sensitive parasites, they displayed increased efficacy compared to CQ against CQ-resistant parasites.

### 2.3. Cytotoxicity and Haemolytic Activity

To evaluate the cytotoxicity and selectivity of the potential anti-infective agents, the PTZ-quinoline hybrids were tested on human-derived fibroblasts (WS1 cell line) and human red blood cells; cytotoxicity assessment in the latter model is widely recognized as providing information regarding the therapeutic value and systemic toxicity of chemicals, drugs, or blood-contacting medical devices or materials [35].

Experimental assays on WS1 cells demonstrated that **4**–**6a**,**b** slightly interfered with cell viability and proliferation, with CC_50_ values in the 11.78–48.27 µM range (Table 2). All the compounds exhibited good selectivity index (SI) values for *P. falciparum*, regardless of the CQ sensitivity of the strain, while lower, but still positive SI values were measured for *S. aureus*. This finding is associated with the overall decreased inhibitory activity (i.e., higher IC_50_ values) of the triazole derivatives against the bacterial strain, which may be attributed to a reduced influx of the compounds into the bacterial cytoplasm due to the cell wall.

All the PTZ-quinoline hybrids were also assayed against human erythrocytes to evaluate their detrimental effect on the structure of the red blood cell membrane. Remarkably, none of the compounds showed haemolytic activity (Appendix A), revealing good biocompatibility and indicating an overall safety of these PTZ-quinoline hybrids, even at the highest concentrations.

Considering the results obtained in these biological assays, and taking antiprotozoal and antimicrobial activities into consideration, the PTZ-quinoline hybrid compounds **4**–**6a**,**b** exhibited the most significant selective effect towards the tested pathogens over host cells, with the exception of derivative **6a**, which did not exhibit antibacterial activity within the tested concentration range.

### 2.4. In Silico Study on NDH-2

Provided the availability of the X-ray crystallographic three-dimensional structures of *S. aureus* and *P. falciparum* NDH-2 isoforms in the protein data bank database [36], we could investigate in more detail the hypothesized mechanism of action, related to the presence in the designed ligands of both the 7-chloro-4-amino-quinoline moiety and the PTZ core. Beforehand, **1**–**8** were accurately modelled to predict their physico-chemical properties employing in silico simulations. Such features were vital to explain the observed antimicrobial and antiprotozoal activities, supported by previous studies on the influence of pH and ionic concentration on NDH-2 turnover and enzymatic activity [37].

First, membrane permeability was measured by computing the logarithm of the RRCK permeability (in cm/s). A conformational search was conducted with an associated probability of the molecule trespassing onto the membrane (low dielectric continuum) following desolvation in water (high dielectric continuum). Obtained membrane permeabilities ranged from 3.2 × 10^−6^ cm/s to 1.3 × 10^−5^ cm/s. Excluding **7** (which represented an outlier lying outside the observed distribution), the most potent anti-staphylococcal triazole derivatives **4**–**5a**,**b** were the most membrane permeable of the series, ranging from 5.7·× 10^−6^ (**5b**) to 5.5·× 10^−6^ cm/s (**4a**) (Figure 3A). All the other compounds demonstrated lower membrane permeability, except for **6a** (active only against *P. falciparum*), which showed a comparable value within distribution (5.5·× 10^−6^ cm/s). It is worth mentioning that membrane permeability was influenced by the most probable protonation states of each molecule. For this reason, in order to predict acid dissociation constants, protonation states accessible at physiological pH were assigned to the molecules. All compounds demonstrated the capability of becoming protonated at the quinoline pyridine nitrogen atom N1, except for molecule **7**, lacking a suitable 4-amino moiety linking the quinoline to the spacer. Therefore, acid dissociation constants (pKas) were computed to quantify the abundance of a protonation state over another. pKas were variably spanned across the range 5.15–6.21, but no significant trend with the calculated IC_50_ values was reported (Figure 3B).

On the other hand, it should be highlighted that single-point charge conformers could be essential to trigger the observed antibacterial and antiprotozoal properties, as the uncharged compound **7** resulted in being completely inactive. Notably, derivatives **1**–**3** and **8** could be simultaneously protonated at the tertiary amine atoms at physiological pH, leading to completely different electrostatic profiles.

In addition, the strictly related redox potential between the uncharged state and the one-electron reduced form of each compound was investigated, as it is closely involved in the enzymatic mechanism and the explanation of the route of electrons exchanged at the protein active site. Ab initio computations implementing B3LYP/6-31G(d) density functional theory were performed in acetonitrile solvent, a robust and widely used procedure for carrying out such quantomechanical (QM) simulations [38,39]. Obtained values ranged from −2.44 to −1.54 V, demonstrating a negative correlation of −0.67 and −0.63 with previously computed IC_50_ values (relative to *S. aureus* ATCC 25923 and *P. falciparum* CQ-S strains, respectively) (Figure 4). This result reinforced our findings, as it confirmed that compounds with a higher (but negative) redox potential could be implicated in a higher antimicrobial activity. In other words, the designed compounds showed a greater tendency overall to become oxidized than reduced, although most active derivatives were closer to −1.0 V, making their corresponding reduced forms accessible in solution. Nevertheless, pKa could be strictly correlated with the redox potential, as the protonation and electron acceptor sites most likely resided on the quinoline N1 atom.

Starting from such a full evaluation of the compounds’ chemical and physical properties, we performed molecular docking simulations, focusing on the binding sites that were previously characterized. The simulations were carried out in parallel on *S. aureus* and *P. falciparum* NDH-2 protein isoforms, starting from each corresponding 3D structure available (PDB: 5NA1 at 2.3 Å resolution [40] and PDB: 5JWC at 2.0 Å resolution [41], respectively). Evidently, the folding was conserved (root mean square deviation on backbone alpha carbons was 1.947 Å), but the aminoacidic sequence (and particularly affecting the binding site) changed significantly between the two protein isoforms (21.72% sequence identity) (Appendix A). Such an outcome might result in much different interaction networks, making it impossible to define a common mechanism a priori, but requiring a focused investigation for both microorganisms. In both cases, three putative binding sites were evaluated: the region occupied by the FAD cofactor, the cytosol accessible cavity of NAD^+^/H, and the membrane-enclosed quinone (Q) binding site (Figure 5). For the last one, HQNO, being previously reported as a *pf*NDH-2 inhibitor [42], served as a reference ligand to investigate the region occupied by the physiological quinone substrate. While the compounds showed an overall good binding affinity towards all these binding sites for both microorganisms (−6 kcal/mol docking score or higher), they demonstrated an even higher binding affinity when compared to the reference ligand within the Q binding site (Appendix A). Concerning *S. aureus*, the most active compound **4b** was reported as the derivative with the lowest root mean square deviation (RMSD) with respect to the reference co-crystallized pose of HQNO (2.36 Å), denoting the most promising interaction network overall (Figure 5). Interestingly, all compounds demonstrated the capability to form hydrogen bonds with the Arg385 sidechain, sometimes reinforced by π–π cation interactions, involving the triazole ring and the PTZ aromatic core (Figure 6). Moreover, the 7-Cl-4-amino-quinoline moiety was crucial to establish halogen bonds to Val353 backbone atoms. **4b** and **5b** were able to reinforce such halogen bond interaction by additionally involving the Lys379 sidechain. Derivatives **5b** and **4a** involved the protonated N1 nitrogen atom of the quinoline group in stabilizing salt bridges with Glu46 and Asp383 sidechains. The Thr48 sidechain established hydrogen bonds towards the same N1-quinoline atom for both **4a** and **5a**. At the same time, compound **5a** demonstrated the unique capability of π–π stacking with Tyr15 headed towards its quinolinic bicyclic core. Lastly, **5a** and **4b** were able to orient their PTZ portion to a π–π stack with the aromatic side ring of Phe366. Interestingly, the identified energy-favourable binding poses of the derivatives lay within the same Q binding pocket of *P. falciparum* NDH-2 compared to *S. aureus*, but the triazole derivatives **4**–**5a**,**b** tended to occupy a larger cleft, potentially impairing the NAD^+^/H and quinone entrance in their usual locations at the same time. In this case, the lowest energy docking scores among the full dataset of compounds were those relative to **4b** and **5b**, being the most active molecules against *P. falciparum* after experimental validation (Appendix A). All compounds, except for **4b**, were able to establish interactions with Phe504, either by π–π stacking (**5b** and **4a**) or by hydrogen bonding (**5a**, **6a**, and **6b**) at different levels, depending on the linker size and orientation (Figure 5B,C, and Figure 6). Nevertheless, the designed compounds showed an irregular interaction network, with fewer points in common. A Trp50 sidechain was involved in π–π stacking or π-cation interactions with the quinoline portion of **5a** and **4a**. In addition, the PTZ moiety of derivatives **4a** and **6a** is oriented to π–π stack with the Phe499 side ring. As standalone cases, **5a** established a halogen bond with Lys470 and **4b** a single hydrogen bond between the Leu473 backbone and the compound’s triazole ring. In particular, the distinct subcellular localization of the NDH-2 enzyme may offer further evidence supporting the significant inhibition of *P. falciparum* cell growth. This difference likely results from substantial hindrance to accessing the enzyme’s binding sites when targeting *S. aureus*. Whether by diffusion through the plasma membrane or by crossing the membrane directly, such interference could prevent NAD^+^/H from entering the binding site from the cytosol. In fact, the compounds could reach the NAD^+^/H binding site of the enzyme directly from the cytosol, while demonstrating comparable or more favourable binding scores than NAD^+^ itself (Appendix A). The docking scores obtained on the FAD binding sites (both for *S. aureus* and *P. falciparum*) were less encouraging if compared to the reference FAD pose, which is non-covalently but tightly bound to the cavity, therefore being much more difficult to displace (Appendix A).

## 3. Discussion

In this paper, the ‘covalent biotherapy’ strategy has been applied by linking two “privileged” molecular scaffolds, namely quinoline and PTZ, endowed with recognized antimalarial and antimicrobial potential, with the aim of expanding the activity profile and possibly engaging the NDH-2 enzymatic complex, a key molecular target selectively expressed in microbial organisms, such as *P. falciparum* and *S. aureus*, but not in mammals. Looking at the antimicrobial results, it could be noticed that the new derivatives showed a selective activity profile towards the Gram-positive *S. aureus*, since no activity was recorded against *E. coli* and *C. albicans*. In particular, compounds **4a**,**b** and **5a**,**b** showed IC_50_ values in the low micromolar range, being the chloro-PTZ derivatives **4b** (IC_50_ = 4.86 µM) and **5b** (IC_50_ = 5.03 µM) slightly more potent than the corresponding unsubstituted PTZ analogues **4a** and **5a** (IC_50_ = 7.49 µM and 6.73 µM, respectively). Despite the very similar IC_50_ values of **5b** and **4b**, the latter, bearing a shorter chain connecting the chloro-PTZ-triazole with the 7-chloroquinoline moiety, appeared the most active. The role of the chain length was confirmed by the further decrease in potency observed with **6b** (IC_50_ = 25.05 µM) bearing a three-carbon atom linker, leading to speculation on the need for a well-defined distance between the two main pharmacophores (Figure 7). The observed anti-staphylococcal activity of **6b** could be due to its fitting within the Q binding site, as suggested by the calculated excellent affinity (−11.174 kcal/mol). However, its non-optimal pharmacokinetic properties hamper the ability to reach the hydrophobic cleft, thus lowering its potency compared to compounds **4**–**5a**,**b**. Notably, the removal of a spacer between the main scaffolds, as in compound **7**, led to a complete loss of antimicrobial activity (Table 1).

The effectiveness of the most potent anti-staphylococcal PTZ-quinoline hybrids **4**–**5a**,**b** was evaluated against MRSA isolates; notably, the compounds proved to be active regardless of their antibiotic resistance profile, with IC_50_ values close to the values measured for the *S. aureus* control. This is clinically relevant considering that isolates may present phenotypic and genetic heterogeneity compared to laboratory reference strains, thus some diversity in susceptibility may occur.

Shifting the focus to antiplasmodial results, all CQ hybrids, except for **7**, showed submicromolar inhibitory activities on both CQ-sensitive (D10) and CQ-resistant (W2) strains of *P. falciparum*, while the PPQ derivatives **1**–**3** and **8** showed a 10-fold decrease in potency compared to the CQ-related compounds. These data are consistent with the measured values of membrane permeability that indicated **4**–**5a**,**b** and **6a** as the most permeable compounds in the series. The ability to cross the membranes is related to the protonation state of the molecule at physiological pH, and, even if no significant trend could be observed between the pKas of the molecules and their corresponding IC_50_ values, it is noteworthy that the inactive **7** was the only compound unable to be protonated, lacking an aminoalkyl chain between the quinoline and the triazole moieties. Moreover, its inactivity on both *S. aureus* and *P. falciparum* highlighted the crucial role of a single-point charge in triggering the observed antibacterial and antiprotozoal activities. On the other hand, the less potent PPQ derivatives **1**–**3** and **8**, showing an overall increased basicity due to the presence of a piperazine core, exhibit a completely different electrostatics profile. As a consequence, these derivatives showed a marked decrease in membrane permeability, as well as a lowered redox potential, demonstrating a strong preference for the oxidized over the one-electron reduced forms. In addition, compounds **1**–**3** and **8** frequently showed less favourable scores on the binding sites in molecular docking studies, especially when considering the *P. falciparum* NDH-2 isoform. At the same time, when docking scores were comparable to the other compounds of the series, or higher in some cases, their poses and established interaction networks differed significantly from those reported for the reference substrates.

Interestingly, almost all compounds were more potent than CQ against CQ-resistant parasites, and similar IC_50_ values against D10 and W2 strains were recorded, indicating the lack of cross-resistance with CQ itself. This evidence seemed to confirm that a different mechanism of action, as well as a different target, may be involved in the antimalarial activity of the new hybrid compounds. Therefore, for both *P. falciparum* and *S. aureus*, molecular docking simulations were performed on the three main putative binding sites of the membrane-bound NDH-2 enzymatic complex, namely the area occupied by the FAD cofactor, the cytosol accessible cavity of NAD^+^/H and the binding site for the Q, and the results showed the ability of the most potent compounds **4**–**5a**,**b** to appreciably interact with all of them, in particular with the Q binding site. In fact, these compounds appeared to be able to occupy a larger cleft when binding to the *pf*NDH-2 isoform, leading to a simultaneous impairment in the binding of Q and NAD^+^/H, with the lowest energy docking scores for **4b** and **5b**, among the most potent derivatives of the series.

Undoubtedly, looking at the results as a whole, a remarkably higher potency of the new compounds towards *P. falciparum* compared to *S. aureus* was observed. These experimental findings can be explained by considering the different cellular architecture of the two microorganisms. In *S. aureus*, the enzyme is embodied in the cytoplasmic membrane that is, in turn, surrounded by a complex cell wall, possibly limiting the passage of the compounds into the cytosol. Furthermore, to reach the Q binding site, the compounds should cross the membrane, which could represent an additional obstacle, hindering the antimicrobial activity. Indeed, the inability of molecules to penetrate and efficiently reach their site of action is a common reason accounting for their low inhibitory potential and the failure of several antibacterial drug discovery projects. On the other hand, when considering *P. falciparum*, NDH-2 is associated with the outward side of the inner mitochondrial membrane, thus accessible to compounds following their diffusion through the sole plasma membrane: the compounds could also reach the NAD^+^/H binding site of the enzyme from the cytosol, thereby preventing NADH from binding. These peculiar features could justify the prevailing activity of the designed compounds on *P. falciparum* with respect to *S. aureus*. Nevertheless, considering that all the most active compounds showed an acceptable selectivity index with respect to mammalian cells, these data provided convincing evidence of the possible application of the STOP strategy to combat malaria co-infections.

## 4. Materials and Methods

### 4.1. Chemistry

#### 4.1.1. General Methods

All chemicals were purchased from Aldrich Chemistry, Milan (Italy), or Alfa Aesar, Milan (Italy), and were of the highest purity. The selected solvents were of analytical grade. Thin-layer chromatography (TLC) on precoated silica gel plates (Merck Silica Gel 60 F254, purchased by Aldrich Chemistry) was used to monitor the reaction progress and then visualized under UV254 lamplight. Compounds purifications were performed by flash chromatography on silica gel columns (Kieselgel 40, 0.040–0.063 mm, Merck, Aldrich Chemistry). Melting points were determined in open glass capillaries, using a Büchi apparatus, and are uncorrected. ^1^H NMR spectra were recorded for the intermediate compounds on a Varian Gemini spectrometer (Agilent technologies, Cernusco sul Naviglio, Milan, Italy) working at 400 MHz, while for the final compounds ^1^H NMR and ^13^C NMR spectra were recorded on a Bruker spectrometer (Bruker Italia S.r.l., Milano, Italy) working at 600 MHz and at 150 MHz, respectively, in CDCl_3_ solutions unless otherwise indicated. Chemical shifts (δ) were reported as parts per million (ppm) values relative to tetramethylsilane (TMS) as internal standard; coupling constants (*J*) are reported in Hertz (Hz). Standard abbreviations were used for indicating spin multiplicities: s (singlet), d (doublet), dd (double doublet), t (triplet), br (broad), q (quartet), or m (multiplet). UHPLC–MS analyses were run on a Waters ACQUITY ARC UHPLC/MS system (Waters SpA, Sesto San Giovanni, Milan, Italy) consisting of a QDa mass spectrometer equipped with an electrospray ionization interface and a 2489 UV/Vis detector at wavelengths (λ) 254 nm and 365 nm. The analyses were performed on an XBridge BEH C18 column (10 × 2.1 mm i.d., particle size 2.5 μm) with a XBridge BEH C18 VanGuard Cartridge precolumn (5 mm × 2.1 mm i.d., particle size 1.8 μm), with mobile phases consisting of H_2_O (0.1% formic acid) (A) and MeCN (0.1% formic acid) (B). Electrospray (ES) ionization in positive mode was applied in the mass scan range of 50–1200 Da. Method and gradients used were the following: Generic method. Linear gradient: 0–0.78 min, 20% B; 0.78–2.87 min, 20–95% B; 2.87–3.54 min, 95% B; 3.54–3.65 min, 95–20% B; 3.65-5-73, 20% B. Flow rate: 0.8 mL/min. All tested compounds were found to have >95% purity. Compounds were named relying on the naming algorithm developed by CambridgeSoft Corporation (Cambridge, USA) and used in ChemBioDraw Ultra (version 23.0).

#### 4.1.2. General Method for the Preparation of N-Bromoalkyl-Phenothiazine Derivatives **9**–**11**

Phenothiazine (1 eq) was dissolved in DMSO (10 mL), and KOtBu (1 eq) was added with stirring. After 1 h, this solution was added dropwise to a solution of the appropriate 1-Br-ωhalo-alkane (1-bromo-3-chloropropane, 1,4-dibromobutane, or 1,5-dibromopentane, 2 eq) dissolved in 10 mL of DMSO. The reaction mixture was stirred at room temperature and followed by TLC until the starting material disappeared, quenched with ice and water, extracted with dichloromethane (DCM), washed with water and brine, and the solvent was evaporated to dryness under reduced pressure. The crude was purified by flash column chromatography.

**10-(3-chloropropyl)-10H-phenothiazine (9).** Using the previous procedure and starting from 1.0 g of phenothiazine (0.005 mol) and 1-bromo-3-chloropropane, 0.85 g of **9** was obtained (PE/EtOAc 4.9:0.1). Yield: 61%, oil. ^1^H NMR δ 2.23–2.28 (m, 2H, -CH_2_-), 3.68 (t, *J* = 6.4 Hz, 2H, N-CH_2_-), 4.10 (t, *J* = 6.4 Hz, 2H, -CH_2_-Cl), 6.91–6.97 (m, 4H, Ar), 7.16–7.20 (m, 4H, Ar).**10-(4-bromobutyl)-10H-phenothiazine (10).** Using the previous procedure and starting from 1.0 g of phenothiazine (0.005 mol) and 1,4-dibromobutane, 0.80 g of **10** was obtained (PE/EtOAc 4.75:0.25). Yield: 48%, oil. ^1^H NMR δ 2.08–2.10 (m, 4H, -CH_2_-CH_2_-), 3.51 (t, *J* = 6.0 Hz, 2H, N-CH_2_-), 4.02 (t, *J* = 6.0 Hz, 2H, -CH_2_-Br), 6.97–7.06 (m, 4H, Ar), 7.26–7.29 (m, 4H, Ar).**10-(5-bromopentyl)-10H-phenothiazine (11).** Using the previous procedure and starting from 1.0 g of phenothiazine (0.005 mol) and 1,5 dibromopentane, 0.90 g of **11** was obtained (PE/EtOAc 4.9:0.1). Yield: 52%, oil. ^1^H NMR δ 1.56–1.60 (m, 2H, -CH_2_-), 1.80–1.88 (m, 4H, -CH_2_-CH_2_-), 3.36 (t, *J* = 6.4 Hz, 2H, N-CH_2_-), 3.86 (t, *J* = 6.4 Hz, 2H, -CH_2_-Br), 6.83–6.92 (m, 4H, Ar), 7.11–7.15 (m, 4H, Ar).**10-(3-(4-(7-chloroquinolin-4-yl)piperazin-1-yl)propyl)-10H-phenothiazine (1).** In a Schlenk tube, **9** (0.35 g, 1.3 mmol, 1 eq), 7-chloro-4-(piperazin-1-yl)quinoline (0.47 g, 1.9 mmol, 1.5 eq), and triethylamine (TEA) (0.19 g, 0.26 mL, 1.5 eq) were dissolved in acetonitrile. The reaction mixture was heated at 70 °C for 24 h. The solvent was removed under reduced pressure, and the crude product was purified by flash column chromatography (PE/EtOAc 4.75:0.25) to yield 0.12 g of **1** (yield 20%), m.p. 72–73 °C. ^1^H NMR δ 2.18–2.21 (m, 2H, -CH_2_-), 2.85–2.91 (m, 6H, N-CH_2_- + piperazine), 3.37 (s, 4H, piperazine), 4.05 (t, *J* = 6.0 Hz, 2H, -CH_2_-N-PTZ), 6.84 (d, *J* = 6.0 Hz, 1H, Ar), 6.92–6.97 (m, 4H, Ar), 7.17–7.20 (m, 4H, Ar), 7.42 (d, *J* = 12.0 Hz, 1H, Ar), 7.84 (d, *J* = 12.0 Hz, 1H, Ar), 8.09 (s, 1H, Ar), 8.71 (d, *J* = 6.0 Hz, 1H, Ar). ^13^C NMR: δ 45.1, 53.1, 55.6, 109.1, 115.8, 121.9, 122.8, 125.2, 125.5, 126.4, 127.4, 127.7, 128.9, 135.2, 145.3. MS (ES) *m*/*z*: 488.1 (M + H).**10-(4-(4-(7-chloroquinolin-4-yl)piperazin-1-yl)butyl)-10H-phenothiazine (2).** 0.36 g of **10** (1.1 mmol, 1.0 eq), 0.27 g of 7-chloro-4-(piperazin-1-yl)quinoline (1.1 mmol, 1.0 eq), and 0.11 g (0.16 mL, 1.0 eq) of TEA were dissolved in toluene. The reaction mixture was heated under reflux for 8 h, cooled at r.t., washed with H_2_O, dried over anhydrous Na_2_SO_4,_ and the solvent evaporated to dryness. The crude was purified by flash column chromatography (PE/EtOAc (NH_4_OH 5%) 8:2) to yield 0.15 g of **2** (yield 27%), m.p. 78–80 °C. ^1^H NMR: δ 1.74–1.93 (m, 4H, -CH_2_CH_2_-), 2.54 (s, 2H, N-CH_2_), 2.67–2.71 (m, 4H, piperazine), 3.18–3.21 (m, 4H, piperazine), 3.94 (t, *J* = 6.0 Hz, 2H, CH_2_-N-PTZ), 6.81 (s, 1H, Ar), 6.89–6.94 (m, 4H, Ar), 7.14–7.17 (m, 4H, Ar), 7.40 (d, *J* = 12.0 Hz, 1H, Ar), 7.90 (d, *J* = 12.0 Hz, 1H, Ar), 8.04 (s, 1H, Ar), 8.71 (d, *J* = 6.0 Hz, 1H,Ar). ^13^C NMR: δ 23.5, 24.6, 47.0, 52.0, 52.9, 57.8, 109.1, 115.8, 122.0, 122.7, 125.2, 125.4, 126.3, 127.4, 127.7, 129.0, 135.1, 145.4, 150.2, 152.0. MS (ES) *m*/*z*: 502.1 (M + H).**10-(5-(4-(7-chloroquinolin-4-yl)piperazin-1-yl)pentyl)-10H-phenothiazine (3).** Applying the procedure used for **2** and starting from 0.66 g (1.9 mmol, 1.0 eq) of **11**, compound **3** was obtained. The crude was purified by flash column chromatography (PE/EtOAc (NH_4_OH 5%) 9:1) to obtain 0.30 g of **3** (yield 53%), m.p. 77–79 °C. ^1^H NMR: δ 1.58–1.48 (m, 4H, CH_2_CH_2_), 1.83–1.88 (m, 2H, CH_2_), 2.44 (t, *J* = 12Hz, 2H, N-CH_2_), 2.68–2.72 (m, 4H, piperazine), 3.19–3.21 (m, 4H, piperazine),, 3.89 (t, *J* = 6.0 Hz, 2H, CH_2_N-PTZ), 6.81–6.92 (m, 5H, Ar), 7.13–7.16 (m, 4H, Ar), 7.41 (d, *J* = 12.0 Hz, 1H, Ar), 7.92 (d, *J* = 12.0 Hz, 1H, Ar), 8.03 (s, 1H, Ar), 8.70 (d, *J* = 6.0 Hz, 1H, Ar). ^13^C NMR: δ 24.9, 26.6, 26.9, 47.2, 52.3, 55.2, 58.5, 109.1, 115.6, 122.0, 122.6, 125.4, 126.2, 127.3, 127.6, 129.0, 135.0, 145.5, 150.3, 152.1, 157.1. MS (ES) *m*/*z*: 515.1 (M + H).**7-chloro-4-(4-(3-chloropropyl)piperazin-1-yl)quinoline (13).** A solution of 7-chloro-4-(piperazin-1-yl)quinoline (1.16 g, 4.7 mmol, 1 eq), 1-bromo-3-chloropropane (0.93 mL, 9.4 mmol, 2 eq), and TEA (0.7 mL, 4.7 mmol, 1 eq) in toluene (20 mL) was heated for 6 h under reflux. The mixture was cooled at r. t., diluted with DCM, washed with H_2_O, dried over anhydrous Na_2_SO_4,_ and the solvent was evaporated to dryness, obtaining 1.00 g of **13** (66%), used for the subsequent step without further purification. ^1^H NMR: δ 1.97–2.04 (m, 2H, CH_2_-CH_2_-CH_2_), 2.59–2.65 (m, 2H, N-CH_2_-), 2.67–2.72 (m, 4H, piperazine), 3.18–3.22 (m, 4H, piperazine), 3.62–3.69 (m, 2H, -CH_2_Cl), 6.78–6.84 (m, 1H, Ar), 7.56 (d, *J* = 16.0 Hz, 1H, Ar), 7.87–7.93 (m, 1H, Ar), 8.03 (s, 1H, Ar); 8.69 (d, *J* = 4.0 Hz, 1H, Ar). MS (ES) *m*/*z*: 324.2 (M + H).**4-(4-(3-azidopropyl)piperazin-1-yl)-7-chloroquinoline (14).** A mixture of 1.0 g (3.1 mmol) of **13** and 0.3 g (4.6 mmol) of NaN_3_ in 25 mL of DMSO was stirred at r.t. for 24 h, then quenched in ice and water, extracted with EtOAc, washed with H_2_O, dried over anhydrous Na_2_SO_4_ and the solvent was evaporated under vacuum to obtain 0.62 g (60%) of **14** used without purification. ^1^H NMR: δ 1.81–1.88 (m, 2H, CH_2_-CH_2_-CH_2_), 2.61 (t, *J* = 12.0 Hz, 2H, N-CH_2_), 2.69–2.75 (m, 4H, piperazine), 3.20–3.25 (m, 4H, piperazine), 3.53 (dt, *J*_1_ = 10.4 Hz, *J_2_* = 12 Hz, 2H, CH_2_N_3_), 6.84 (d, *J* = 8.0 Hz, 1H, Ar), 7.42 (dd, *J*_1_ = 4.0 Hz, *J_2_* = 8.0 Hz, 1H, Ar), 7.95 (d, *J* = 8.0 Hz, 1H, Ar), 8.05 (d, *J* = 4.0 Hz, 1H, Ar), 8.72 (d, *J* = 8.0 Hz, 1H, Ar). MS (ES) *m*/*z*: 331.2 (M + H).**2-(5-azidopentyl)isoindoline-1,3-dione (15).** Applying the procedure used for **14** and starting from 0.5 g (1.7 mmol) of 2-(5-bromopentyl)isoindoline-1,3-dione and 0.16 g (2.5 mmol) of NaN_3_, 0.4 g of **15** was obtained (91%), as an oily compound. ^1^H NMR: δ 1.41–1.45 (m, 2H), 1.58–1.74 (m, 4H), 3.28 (t, *J* = 12.0 Hz, 2H, N-CH_2_), 3.70 (t, *J* = 12.0 Hz, 2H, CH_2_-N), 7.71–7.73 (m, 2H, Ar), 7.84–7.86 (m, 2H, Ar). MS (ES) *m*/*z*: 259.3 (M + H).**General procedures for the click chemistry CuAAC reaction (4–5a,b, 7, 8, 16a,b).** To a solution of the alkyne (1 eq) in DMSO, the selected azide (1.3 eq) and TEA (4.7 eq) were added. A solution of CuSO_4_ (0.1 eq) and sodium ascorbate (0.5 eq) in H_2_O was prepared and promptly added to the reaction mixture, which was stirred for 72 h at r.t. and then poured into ice. The mixture was extracted with EtOAc (3 × 40 mL), and the organic layer was washed with H_2_O, dried over Na_2_SO_4,_ and evaporated to dryness. The obtained crude product was purified by flash chromatography or crystallization.***N*-(2-(4-((10H-phenothiazin-10-yl)methyl)-1H-1,2,3-triazol-1-yl)ethyl)-7-chloroquinolin-4-amine (4a).** Using the previous procedure, 0.17 g (0.7 mmol) of *N*-(2-azidoethyl)-7-chloroquinolin-4-amine [28] was reacted with 0.13 g (0.53 mmol) of **12a**. The crude material was purified by flash chromatography (DCM/MeOH 9.75:0.25), to obtain 0.07 g of **4a** (yield 27%), m.p. 110–112 °C. ^1^H NMR: δ 3.78–3.81 (m, 2H, -CH_2_NH), 4.62 (t, *J* = 6.4 Hz, 2H, -CH_2_-N), 5.20 (s, 2H, -CH_2_-NPhen), 5.58 (broad, 1H, NH), 6.35 (s, 1H, Ar), 6.70 (d, *J* = 6.4 Hz, 2H, Ar), 6.86 (t, *J* = 6.4 Hz, 2H, Ar), 6.95 (t, *J* = 12.0 Hz, 2H, Ar), 7.10 (d, *J* = 12.0 Hz, 2H, Ar), 7.35 (s, 1H, Ar), 7.28 (dd, *J*_1_ = 12.0 Hz, *J_2_* = 6.0 Hz, 1H, Ar), 7.56 (d, *J* = 6.0 Hz, 1H, Ar), 7.97 (s, 1H, triazole), 8.52 (d, *J* = 6.0 Hz, 1H, Ar). ^13^C NMR: δ 42.9, 44.9, 49.0, 99.0, 115.2, 117.3, 121.0, 123.1, 123.5, 124.1, 126.2, 127.4, 127.5, 129.0, 135.5, 144.2, 145.6, 149.0, 149.2, 152.0. MS (ES) *m*/*z*: 486.0 (M + H).**7-chloro-*N*-(2-(4-((2-chloro-10H-phenothiazin-10-yl)methyl)-1H-1,2,3-triazol-1-yl)ethyl)quinolin-4-amine (4b).** Using the previous procedure, 0.21 g (0.85 mmol) of *N*-(2-azidoethyl)-7-chloroquinolin-4-amine [28] was reacted with 0.18 g (0.65 mmol) of **12b**. The crude was purified by flash chromatography (EtOAc/EP 9:1), to obtain 0.03 g of **4b** (yield 9%), m.p. 104–106 °C. ^1^H NMR (DMSO-*d_6_*): δ 3.72–3.75 (m, 2H, -CH_2_NH), 4.64 (t, *J* = 12.0 Hz, 2H, -CH_2_-N), 5.09 (s, 2H, -CH_2_-N-PTZ), 6.47 (d, *J* = 6.4 Hz, 1H, Ar), 6.84 (d, *J* = 12 Hz, 1H, Ar), 6.88–6.92 (m, 3H, Ar), 6.99 (t, *J* = 18.0 Hz, 1H, Ar), 7.06–7.09 (m, 2H, Ar), 7.37 (br, 1H, NH), 7.41 (dd, *J*_1_ = 12.0 Hz, *J*_2_ = 3.0 Hz, 1H, Ar), 7.78 (d, *J* = 3.0 Hz, 1H, Ar), 8.02 (s, 1H, triazole), 8.09 (d, *J* = 12.0 Hz, 1H, Ar), 8.36 (d, *J* = 6.0 Hz, 1H, Ar). ^13^C NMR: δ 42.4, 44.1, 48.0, 98.8, 115.5, 115.9, 117.4, 121.3, 122.0, 122.2, 123.0, 123.8, 123.9, 124.3, 126.7, 127.4, 127.5, 132.2, 133.0, 142.9, 143.3, 145.4, 149.7, 151.8. MS (ES) *m*/*z*: 519.2 (M + H).***N*-(3-(4-((10H-phenothiazin-10-yl)methyl)-1H-1,2,3-triazol-1-yl)propyl)-7-chloroquinolin-4-amine (5a).** Using the previous procedure, 0.17 g (0.65 mmol) of *N*-(3-azidopropyl)-7-chloroquinolin-4-amine [28] was reacted with 0.12 g (0.5 mmol) of **12a**. The crude was purified by flash chromatography (EtOAc/EP 9:1), to obtain 0.08 g of **5a** (yield 32%), m.p. 98–100 °C. ^1^H NMR (Acetone-d): δ 2.31–2.36 (m, 2H, -CH_2_-), 3.36–3.38 (m, 2H, -CH_2_NH-), 4.57–4.60 (m, 2H, -CH_2_N); 5.17 (s, 2H, -CH_2_N-PTZ); 6.42 (d, *J* = 12.0 Hz, 1H, Ar); 6.63 (br, 1H, NH); 6.89–6.93 (m, 4H, Ar); 7.05–7.11 (m, 4H, Ar); 7.37 (d, *J* = 12.0 Hz, 1H, Ar), 7.80 (s, 1H, triazole), 7.82 (d, *J* = 6.0 Hz, 1H, Ar), 8.09 (d, *J* = 12.0 Hz, 1H, Ar), 8.41 (d, *J* = 6.0 Hz, 1H, Ar). ^13^C NMR (CDCl_3_): δ 28.4, 40.1, 45.0, 48.2, 99.0, 115.3, 117.3, 121.1, 123.1, 124.1, 125.8, 127.5, 129.1, 135.3, 144.3, 145.7, 149.2, 149.3, 152.1. MS (ES) m/s 500.0 (M + H).**7-chloro-*N*-(3-(4-((2-chloro-10H-phenothiazin-10-yl)methyl)-1H-1,2,3-triazol-1-yl)propyl)quinolin-4-amine (5b).** Using the previous procedure, 0.22 g (0.85 mmol) of *N*-(3-azidopropyl)-7-chloroquinolin-4-amine [28] was reacted with 0.18 g (0.65 mmol) of **12b**. The crude was purified by flash chromatography (EtOAc/EP 9:1), to obtain 0.13 g of **5b** (yield 37%), m.p. 106–108 °C. ^1^H NMR: δ 2.26–2.28 (m, 2H, -CH_2_-), 3.30–3.33 (m, 2H, -CH_2_NH-), 4.46 (t, *J* = 6.0 Hz, 2H, -CH_2_N), 5.16 (s, 2H, -CH_2_N-PTZ), 5.31 (br, 1H, NH), 6.32 (d, *J* = 3.0 Hz, 1H, Ar), 6.69 (d, *J* = 6.0 Hz, 1H, Ar), 6.72 (s, 1H, Ar), 6.87–6,91 (m, 2H, Ar), 6.97 (t, *J* = 18.0 Hz, 1H, Ar), 7.01 (d, *J* = 12.0 Hz, 1H, Ar), 7.10 (d, *J* = 6.0 Hz, 1H, Ar), 7.34 (s, 1H, CHtri), 7.39 (d, *J* = 12.0 Hz, 1H, Ar), 7.65 (d, *J* = 12.0 Hz, 1H, Ar), 7.95 (s, 1H, Ar), 8.5 (d, *J* = 3.0 Hz, 1H, Ar). ^13^C NMR: δ 28.5, 39.9, 45.1, 48.1, 99.0, 115.6, 115.7, 117.3, 121.1, 122.6, 122.9, 123.5, 123.8, 124.1, 125.8, 127.4, 127.7, 127.9, 129.0, 133.4, 135.2, 143.6, 145.0, 145.6, 149.3, 152.0. MS (ES) *m*/*z*: 533.2 (M + H).**10-((1-(7-chloroquinolin-4-yl)-1H-1,2,3-triazol-4-yl)methyl)-10H-phenothiazine (7).** Using the previous procedure, 0.5 g (2.10 mmol) of compound **12a** was reacted with 0.43 g (2.74 mmol) of 4-azido-7-chloroquinoline [29]. The crude was purified by flash chromatography (EtOAc/EP 3:7), to obtain 0.2 g of **7** (yield 20%), m.p. 120–122 °C. ^1^H NMR: δ 5.46 (s, 2H, CH_2_N-PTZ), 6.90 (d, *J* = 18.0 Hz, 2H, Ar), 6.98 (t, *J* = 18.0 Hz, 2H, Ar), 7.12 (t, *J* = 18.0 Hz, 2H, Ar), 7.20 (d, *J* = 18.0 Hz, 2H, Ar), 7.42 (d, *J* = 6.0 Hz, 1H, Ar), 7.49 (d, *J* = 18.0 Hz, 1H, Ar), 7.64 (d, *J* = 12.0 Hz, 1H, Ar), 7.78 (s, 1H, CHtri), 8.20 (s, 1H, Ar), 9.00 (d, *J* = 6.0 Hz, 1H, Ar). ^13^C NMR: δ 44.9, 115.6, 116.2, 120.6, 123.3, 124.3, 124.5, 124.8, 127.6, 129.2, 129.6, 137.0, 140.9, 144.4, 146.0, 150.3, 151.5. MS (ES) *m*/*z*: 442.9 (M + H).**10-((1-(3-(4-(7-chloroquinolin-4-yl)piperazin-1-yl)propyl)-1H-1,2,3-triazol-4-yl)methyl)-10H-phenothiazine (8).** Using the previous procedure, 0.62 g (1.90 mmol) of **14** was reacted with 0.34 g (1.45 mmol) of **12a**. The crude material was purified by flash chromatography (DCM/MeOH 9.5:0.5), to obtain 0.26 g of **8** (yield 32%), m.p. 110–112 °C. ^1^H NMR: δ 2.06–2.11 (m, 2H), 2.26 (t, *J* = 6.6 Hz, 2H), 2.52–2.53 (m, 4H), 3.15 (s, 4H), 4.44 (t, *J* = 6.6 Hz, 2H), 5.26 (d, *J* = 0.8 Hz, 2H), 6.82–6.85 (m, 2+1H), 6.93 (t, *J* = 6.6 Hz, 2H), 7.09–7.12 (m, 2H), 7.14–7.16 (m, 2H), 7.42 (d, *J* = 0.8 Hz, 1H), 7.46 (dd, *J* = 8.9, 2.2 Hz, 1H), 7.93 (d, *J* = 9.0 Hz, 1H), 8.08 (d, *J* = 2.1 Hz, 1H), 8.76 (d, *J* = 5.0 Hz, 1H). ^13^C NMR: δ 27.0, 44.9, 47.9, 52.2, 52.8, 54.0, 109.1, 115.5, 122.0, 123.0, 123.4, 124.0, 125.2, 126.3, 127.4, 127.5, 129.0, 135.1, 144.5, 144.6, 150.3, 152.0, 156.9. MS (ES) *m*/*z*: 569.1 (M + H).**2-(5-(4-((10H-phenothiazin-10-yl)methyl)-1H-1,2,3-triazol-1-yl)pentyl)isoindoline-1,3-dione (16a).** Using the previous procedure, 0.4 g (1.56 mmol) of **15** was reacted with 0.28 g (1.20 mmol) of **12a**. The crude material was purified by flash chromatography (Toluene/EtOAc 9:1), to obtain 0.12 g of **16a** (yield 20%), which was characterized by MS and used for the subsequent step without further purification. MS (ES) *m*/*z*: 496.3 (M + H).**2-(5-(4-((2-chloro-10H-phenothiazin-10-yl)methyl)-1H-1,2,3-triazol-1-yl)pentyl)isoindoline-1,3-dione (16b).** Using the previous procedure, 0.4 g (1.56 mmol) of **15** was reacted with 0.33 g (1.20 mmol) of **12b** in 25 mL of DMSO. The crude was purified by flash chromatography (EP/EtOAc 8:2), to obtain 0.13 g of **16b** (yield 21%), which was characterized by MS and used for the subsequent step without further purification. MS (ES) *m*/*z*: 530.3 (M + H).**5-(4-((10H-phenothiazin-10-yl)methyl)-1H-1,2,3-triazol-1-yl)pentan-1-amine (17a).** 0.12 g (0.24 mmol, 1 eq) of **16a** was dissolved in 10 mL of ethanol, and 0.035 mL (0.72 mmol, 3 eq) of hydrazine hydrate was added. The mixture was heated under reflux for 3 h, cooled to r.t., and acidified with 6 drops of HCl 37%. The solution was stirred for 30 min., and the solvent was removed under reduced pressure. The residue was diluted with water, alkalinized with K_2_CO_3,_ and the precipitate was collected by filtration. The crude material was purified by flash chromatography (Toluene/acetone 3:2) to obtain 60 mg of **17a** (yield 68%). ^1^H NMR: δ 1.23–1.29 (m, 2H, -CH_2_-), 1.42–1.45 (m, 2H, -CH_2_-), 1.79–1.85 (m, 2H, -CH_2_-), 2.17–2.22 (m, 2H, -CH_2_-), 2.65 (br, 2H, -NH_2_), 4.26 (t, *J* = 8.0 Hz, 2H, N-CH_2_-), 5.20 (s, 2H, -CH_2_N-PTZ), 6.77 (d, *J* = 8.0 Hz, 2H, Ar), 6.90 (t, *J* = 8.0 Hz, 2H, Ar), 7.05 (t, *J* = 8.0 Hz, 2H, Ar), 7.12 (d, *J* = 8.0 Hz, 2H, Ar), 7.29 (s, 1H, CHtri). MS (ES) *m*/*z*: 366.3 (M + H).**5-(4-((2-chloro-10H-phenothiazin-10-yl)methyl)-1H-1,2,3-triazol-1-yl)pentan-1-amine (17b).** Using the previous procedure, starting from 0.45 g (1.12 mmol) of **16b** and 0.16 mL (3.36 mmol) of hydrazine hydrate, 0.3 g of **17b** were obtained (yield 65%). ^1^H NMR: δ 1.21–1.27 (m, 2H, -CH_2_-), 1.43–1.49 (m, 2H, -CH_2_-), 1.82–1.89 (m, 4H, -CH_2_-CH_2_-), 2.68 (br, 2H, -NH_2_), 4.31 (t, *J* = 8.0 Hz, 2H, N-CH_2_-), 5.18 (s, 2H, -CH_2_N-PTZ), 6.75–6.80 (m, 2H, Ar), 6.87–6.94 (m, 2H, Ar), 6.99–7.27 (m, 3H, Ar), 7.31 (s, 1H, triazole). MS (ES) *m*/*z*: 400.3 (M + H).**N-(5-(4-((10H-phenothiazin-10-yl)methyl)-1H-1,2,3-triazol-1-yl)pentyl)-7-chloroquinolin-4-amine (6a).** 60 mg (0.164 mmol, 1 eq) of **17a** were treated with 0.16 g (0.82 mmol, 5eq) of 4,7-dichloroquinoline, and the mixture was heated for 7 h at 120–130 °C under N_2_. The residue was diluted with water, and the precipitate formed was filtered and purified by flash chromatography (EtOAc/EP 4:1), to obtain 10 mg (yield 12%) of **6a**, m.p. 104–106 °C. ^1^H NMR: δ 1.21–1.25 (m, 2H, -CH_2_-), 1.88–1.95 (m, 2H, -CH_2_-), 1.99–2.07 (m, 2H, -CH_2_-), 3.24–3.29 (m, 2H, -CH_2_-), 4.33 (t, *J* = 6.0 Hz, 2H, -NCH_2_-), 5.22 (s, 2H, -CH_2_N-PTZ), 6.35 (d, *J* = 4.0 Hz, 1H, Ar), 6.77 (d, *J* = 8.0 Hz, 2H, Ar), 6.89 (t, *J* = 8.0 Hz, 2H, Ar), 6.99–7.05 (m, 2H, Ar), 7.13 (d, *J* = 8.0 Hz, 2H, Ar), 7.32 (s, 1H, triazole), 7.38 (d, *J* = 4.0 Hz, 1H, Ar), 7.76 (d, *J* = 8.0 Hz, 1H, Ar); 7.97 (s, 1H, Ar), 8.49 (d, *J* = 4.0 Hz, 1H, Ar). ^13^C NMR: δ 24.1, 27.5, 29.6, 44.1, 45.9, 53.1, 99.3, 116.4, 116.7, 120.1, 121.8, 122.4, 122.8, 123.4, 123.6, 125.9, 126.6, 127.1, 127.9, 143.5, 144.2, 145.3. MS (ES) *m*/*z*: 527.2 (M + H).**7-chloro-N-(5-(4-((2-chloro-10H-phenothiazin-10-yl)methyl)-1H-1,2,3-triazol-1-yl)pentyl)quinolin-4-amine (6b).** Using the previous procedure, starting from 0.3 g (0.75 mmol) of **17b** and 0.74 g (3.76 mmol) of 4,7-dichloroquinoline, 65 mg (yield 15%) of **6b** were obtained, m.p. 111–113 °C. ^1^H NMR: δ 1.33–1.38 (m, 2H, -CH_2_-), 1.71–1.74 (m, 2H, -CH_2_-), 1.92–1.94 (m, 2H, -CH_2_-), 3.24–3.27 (m, 2H, -CH_2_-), 4.35 (t, *J* = 6.0 Hz, 2H, -NCH_2_-), 4.90 (br, 1H, NH), 5.21 (s, 2H, -CH_2_N-PTZ), 6.35 (d, *J* = 6.0 Hz, 1H, Ar), 6.73 (s, 1H, Ar), 6.76 (d, *J* = 6.0 Hz, 1H, Ar), 6.88 (d, *J* = 12.0 Hz, 1H, Ar), 6.91 (t, *J* = 6.0 Hz, 1H, Ar), 7.00–7.04 (m, 2H, Ar), 7.10 (d, *J* = 6.0 Hz, 1H, Ar), 7.31 (s, 1H, triazole), 7.37 (d, *J* = 6.0 Hz, 1H, Ar), 7.68 (d, *J* = 6.0 Hz, 1H, Ar), 7.96 (s, 1H, Ar), 8.52 (d, *J* = 6.0 Hz, 1H, Ar). ^13^C NMR: δ 23.8, 28.05, 29.8, 43.1, 45.3, 50.1, 99.1, 115.6, 115.7, 121.1, 122.5, 122.6, 122.8, 123.5, 123.7, 125.7, 127.4, 127.7, 127.9, 133.4, 143.7, 144.5, 145.6. MS (ES) *m*/*z*: 561.5 (M + H).

### 4.2. Biological Evaluations

#### 4.2.1. PTZ-Quinoline Compounds and Reference Drugs

The dry powder of the PTZ-quinoline hybrid compounds was dissolved in DMSO at a concentration of 20 mM and then diluted with the appropriate cell medium to achieve the required concentrations. The commercially available drugs were purchased from Sigma-Aldrich (St. Louis, MO, USA), dissolved in water, and stored at 4 °C. Gentamicin and ampicillin were used as reference controls in the antimicrobial testing with *S. aureus* and *E. coli*; fluconazole was used in the assays with *C. albicans*; chloroquine was the standard drug in the *P. falciparum* susceptibility assays. Cisplatin was included as a clinical drug control in the cytotoxicity studies with WS1 cells.

#### 4.2.2. Bacterial Strains and Growth Conditions

Reference bacterial strains of *Staphylococcus aureus* (ATCC 25923) and *Escherichia coli* (ATCC 25922) were used in this study as Gram-positive and negative models, respectively. These reference strains were purchased from the American Type Culture Collection (ATCC, Manassas, VA, USA). Clinical isolates, identified by MALDI-TOF MS (Bruker Daltonik, GmbH, Bremen, Germany), were profiled for their antibiotic susceptibility by using the Vitek2 semiautomated system (bioMérieux, Craponne, France) and according to EUCAST (Version 12, 2022, http://www.eucast.org (accessed on 1 January 2022)). The bacterial cultures were routinely grown on 5% blood agar plates (Biolife Italiana S.r.l., Milan, Italy) and were freshly used in the antibacterial assays.

#### 4.2.3. Candida Albicans and Growth Conditions

The reference strain of *Candida albicans* (ATCC 10231) was used in the present study and maintained on Sabouraud Dextrose Agar (Biolife Italiana S.r.l., Milan, Italy) before the antifungal assays.

#### 4.2.4. Parasite Growth and Drug Susceptibility Assay

*P. falciparum* cultures were carried out according to Trager and Jensen with slight modifications [43]. The CQ-sensitive strain D10 and the CQ-resistant strain W2 were maintained at 5% hematocrit (human A-positive erythrocytes) in RPMI 1640 medium (EuroClone S.p.A, Milan, Italy) with the addition of 1% AlbuMax (Invitrogen, San Diego, California, USA), 0.01% hypoxanthine, 20 mM Hepes, and 2 mM glutamine at 37 °C in a standard gas mixture (1% O_2_, 5% CO_2_, and 94% N_2_). All compounds were dissolved in DMSO and then diluted with medium to achieve the required concentrations (final DMSO concentration <1%, non-toxic to the parasite). Drugs were placed in 96-well flat-bottomed microplates, and serial dilutions were made. Asynchronous cultures with parasitaemia of 1–1.5% and 1% final hematocrit were aliquoted into the plates and incubated for 72 h at 37 °C. Parasite growth was determined spectrophotometrically (OD_650nm_) by measuring the activity of the parasite lactate dehydrogenase (pLDH), as described in a modified version of Makler’s method in control- and drug-treated cultures [44]. The antiplasmodial activity is expressed as 50% inhibitory concentrations (IC_50_); each IC_50_ value is the mean ± standard deviation of at least three separate experiments performed in duplicate.

#### 4.2.5. Antimicrobial Activity

The minimum inhibitory concentrations (MICs) were determined using the microbroth dilution method in microtiter plates according to the Clinical and Laboratory Standards Institute (CLSI) guidelines. In short, microbial inocula were prepared at 0.5 McFarland in PBS and, subsequently, bacterial suspensions were diluted 1:200 in Mueller–Hinton broth (Sigma-Aldrich, St. Louis, MO, USA), while fungal inoculum was diluted 1:20 in RPMI-1640 medium (Gibco^®^, ThermoFisher Scientific Inc., Waltham, MA, USA), containing glucose 2%, 0.3% levo-glutamine buffered to pH 7.0 with 0.165 M 3-(N-morpholino)propanesulfonic acid (MOPS). A total of 100 µL of these microbial suspensions was introduced into a 96-well microplate and incubated with 100 µL of the compounds, two-fold serially diluted in the range of 100–0.19 µM. Positive controls (microbial suspensions in regular media), negative controls (only compounds), solvent controls (microbial suspensions incubated with DMSO dilutions), and the reference drug controls were included in the tests. The plate was incubated at 37 °C for 24 h, and subsequently the optical density at 630 nm (OD_630nm_) was spectrophotometrically measured. Experiments were performed in triplicate with three biological replicates. In addition to MIC, IC_50_ values were measured for the active compounds. The values, corresponding to the concentration of the compound giving rise to an inhibition of growth of 50%, were obtained by interpolation on dose–response curves generated by plotting the percentages of growth inhibition, relative to the positive control (set to 100% of growth), as a function of the tested concentrations (on a logarithm scale). Statistical analysis was carried out by using GraphPad Prism version 9.4.1 for Windows (GraphPad Software, San Diego, CA, USA, www.graphpad.com).

#### 4.2.6. Cell Viability and Proliferation Assay

WS1 cells (ATCC CRL-1502) were selected as a model system to investigate the effect of PTZ-quinoline hybrid compounds on human-derived fibroblasts. Briefly, cells were cultured in DMEM-High glucose supplemented with 10% fetal bovine serum (FBS) (Carlo Erba Reagents, Milan, Italy), 100 U/mL penicillin, and 100 µg/mL streptomycin at 37 °C with 5% CO_2_. For experiments, cells were seeded into 96-well plates at 0.7 × 10^4^ cells/well, and incubated at 37 °C for 24 h. Following washes with PBS, the cell monolayer was incubated with 100 µL of medium containing the 2-fold serial dilutions of the compounds, and of the cisplatin in the range 100–0.19 µM. Cell viability was assessed by a WST8-based assay according to the manufacturer’s instructions (CCK-8, Cell Counting Kit-8, Dojindo Molecular Technologies, Rockville, MD, USA). After 48 h of incubation, culture medium was removed from each well, the monolayer was washed with PBS, and 100 µL of fresh medium containing 10 µL of CCK-8 solution was added. Following a 2 h incubation at 37 °C, the OD_450nm_ was read, and data were expressed as the percentage of cell viability relative to the untreated controls. The CC_50_ was obtained on the corresponding dose–response curves generated as previously reported for IC_50_ values. Experiments were carried out in triplicate in three biological replicates.

#### 4.2.7. Haemolytic Activity Assay

The haemolytic activity of the compounds was evaluated by measuring the amount of hemoglobin released by the disruption of human red blood cells (hRBCs) [35]. For the experiments, fresh hRBCs, obtained from the peripheral blood of anonymous blood donors available for research purposes, were collected by centrifugation at 1500× *g* for 10 min, washed 3 times with PBS, and resuspended to a final concentration of 4% *w*/*v* hRBCs in PBS. Then, 100 µL of hRBCs suspension and an equal volume of the 2-fold dilutions (range 100–0.78 µM) of the antimicrobial peptides were mixed in a 96-well plate and incubated for 1 h at 37 °C. After centrifugation at 1000× *g* for 5 min, the supernatants were transferred into a clear 96-well plate, and OD_405nm_ was read. Untreated hRBCs (incubated with PBS) and hRBCs incubated with 1% Triton X-100 were employed as negative and positive controls, respectively. The haemolysis percentage was calculated as [OD_405nm_ (sample) − OD_405nm_ (negative control)]/[OD4_405nm_ (positive control) − OD_405nm_ (negative control)] × 100. Minimal haemolytic concentrations (MHCs) were defined as the compound concentration causing 10% haemolysis. Three independent experiments were performed in triplicate.

### 4.3. Computational Protocols

#### 4.3.1. Ab Initio pKa Computation

A thermodynamic cycle accounting for the Gibbs free energy (ΔG) of the equilibrium between the protonated and unprotonated forms of each molecule was performed in three steps using the pKa Jaguar tool within Schrödinger suites 2022-3 [45]. The geometry optimization of the three-dimensional coordinates was performed using the B3LYP/6-31G* density functional theory, followed by single-point energy calculations with the cc-pVTZ+ basis set and by solvation free energy contributions with the self-consistent reaction field (SCRF).

#### 4.3.2. Physics-Based Membrane Permeability Prediction

Prime module for membrane permeability computation within Schrödinger suites 2022-3 was used to obtain the energetic scores of diffusion and the logarithm of the predicted membrane permeability through the Ralph Russ Canine Kidney cell line (RRCK) in cm/s (log_10_RRCK). A conformational search was performed in both high-(solution) and low-dielectric (membrane) continuums, accounting for variations in energy. At the end, the most probable conformations in a membrane environment were identified. All parameters were set to default.

#### 4.3.3. Redox Potential Computation

The three-dimensional coordinates of the compounds served as the starting point for a geometry optimization step, performing QM simulations within Gaussian 16 software [46]. Such computations were carried out for each compound and its corresponding one-electron-reduced form, according to B3LYP/6-31G(d) density functional theory. Following geometry optimization, frequency or single-point energy calculations of the solvation contribution were conducted. Solvation energy was computed in acetonitrile solvent, employing the polarizable continuum model by Barone and Cossi (CPCM) [47,48]. Tight convergence of the integrals (SCF) was requested at each step. In addition, the united-atom topological model (UA0) of atomic radius was used during the last step, leading to more robust results according to a previously developed method [39]. Free Gibbs energy ΔG_red_ in the gas phase was obtained by subtracting the SCF energies of each compound in the one-electron reduced form from the oxidized form, to which we applied thermal correction terms. Solvation energies (ΔG_solv_) were obtained by subtracting the solvation energy from the computed SCF energies of their corresponding oxidized or one-electron reduced forms in the gas phase. The final ΔG_red_(solv) terms were obtained by simple resolution of a Born-Haber cycle:



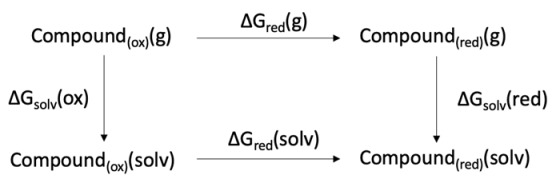



Lastly, the redox potentials were computed as follows:$$E^{0}=\;\frac{{∆G}_{red}(solv)}{-nF}$$
where *F* is the Faraday constant and n is the number of exchanged electrons.

#### 4.3.4. Molecular Modelling and Docking Simulations

NDH-2 protein 3D structure was retrieved from the protein data bank database (PDB) and prepared according to the Protein Preparation Wizard from Schrodinger suites 2022-3 [49]. Hydrogen atoms were removed, re-added, and then minimized according to the protonation of the residues at physiological pH (7.00 ± 2) using Epik [50]. Missing sidechains were modelled using the Prime module. The present procedure was followed either starting from PDB: 5NA1 [40] or PDB: 5JWC [41]. Ligands were manually sketched and then converted into their corresponding three-dimensional coordinates using LigPrep routine. Given the elevated number of atoms and degrees of freedom of the molecules **1**–**8**, a thorough conformational search was performed, using MacroModel. OPLS4 force field [51] was employed alongside the Generalized-Born/Surface-Area (GB/SA) solvation model of water to carry out Montecarlo torsional sampling (MCMM). Extended sampling was selected, where nonbonded interactions cutoffs were set at 8, 20, and 4 Å for Van der Waals, electrostatics, and hydrogen bond interactions, respectively. An energy window of 50 kJ/mol was additionally set to identify new conformers, which were later minimized using the Polak–Ribière Conjugate Gradient algorithm. Conformers with an RMSD of 0.5 Å or less with respect to already selected conformers were discarded. The binding sites were defined using either flavin adenine dinucleotide (FAD), nicotinamide adenine dinucleotide NAD^+^ or 2-heptyl-4-hydroxyquinoline-N-oxide (HQNO) as reference ligands for the Receptor Grid Generation step. NAD^+^ and HQNO coordinates were retrieved after superposition of 5JWB [41] and 6BDO [29] co-crystalized PDB entries to the *S. aureus* and *P. falciparum* NDH-2 model, respectively. All conformers that were generated were fed as input to the Glide molecular docking protocol. During docking, all OH and SH sidechain moieties were allowed to rotate. Glide SP flexible docking protocol was adopted for FAD and NAD^+^/H binding sites, retaining compounds’ input charges and applying strain correction terms to the final docking score [52]. For the Q binding site, it was crucial to utilize the InducedFit protocol to account for the diffusion of the compounds into the “closed” binding site [53]. In this case, the sampling of conformations was carried out on-the-fly, without the aid of previous conformational sampling, which is already a computationally demanding and more accurate task than simple molecular docking simulations. The Glide SP docking function was adopted before and after binding site opening to score the poses.

## Data Availability

The datasets generated and/or analyzed during the current study are available in the AMS Acta repository [https://doi.org/10.6092/unibo/amsacta/8071].

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
