# Peer review of "STOP Strategy to Inhibit P. falciparum and S. aureus Growth: Molecular Mechanism Studies on Purposely Designed Hybrids"

_antibiotics, 2025, doi:10.3390/antibiotics14100991_

Round 1

Reviewer 1 Report

Comments and Suggestions for Authors

Dear Authors,

We congratulate for your great job on the designing and developing antimalarial drug to deal with co-infectious event by inhibiting corresponding microorganisms. By merging two differently active scaffolds against the two factors of malarial and bacterial agents leading to the identified novel compounds, your work confirmed the consistency between the lab-based activities and the computational modelling of the novel compounds. The findings could be serving potential guidance for those who interest in the same topic to design more effective synthesised drugs. Nevertheless, some noticed improper use of terms are found in the manuscript as listed in the given attached file. Thus, improvement is recommended to get the manuscript better in presentation.

Best of luck,
Reviewer

Author Response

Reviewer 1

We congratulate for your great job on the designing and developing antimalarial drug to deal with co-infectious event by inhibiting corresponding microorganisms. By merging two differently active scaffolds against the two factors of malarial and bacterial agents leading to the identified novel compounds, your work confirmed the consistency between the lab-based activities and the computational modelling of the novel compounds. The findings could be serving potential guidance for those who interest in the same topic to design more effective synthesised drugs. Nevertheless, some noticed improper use of terms are found in the manuscript as listed in the given attached file. Thus, improvement is recommended to get the manuscript better in presentation.

We thank the reviewer for the positive feedback and appreciation of our work. We have carefully reviewed and corrected all errors reported in the pdf file.

Reviewer 2 Report

Comments and Suggestions for Authors

The article "STOP strategy to inhibit P. falciparum and S. aureus growth: molecular mechanism studies on purposely designed hybrids," has been designed and written very elegantly. The author has recommended making some changes based on the suggestions highlighted in the PDF file. Second, if appropriate, please put all NMR data in a table in the materials and methods section. It seems like too many words and reading.

Comments on the Quality of English Language

The quality of English seems excellent. The author has suggested reconsidering the discussion part.

Reviewer 3 Report

Comments and Suggestions for Authors
  1. Compound 1 proton NMR is not ok. The integrals written do not match the spectra in the aliphatic region. Please check other proton NMR also.
  2. Mass analysis does not include whether it was in positive ion mode or negative ion mode.
  3. Please include melting points of the final compounds. These are important for analyzing the purity of the compounds.
  4. Please provide a reference for this line (First line of second paragraph on page number 3) "The phenothiazine (PTZ) core, typically present in antipsychotic drugs".
  5. Provide the reference for hybrid chemistry.
  6. The 1,2,3 triazole-containing derivatives 4-5a-b, please include a-b inside brackets (on page 5, first paragraph).
  7. It is a suggestion to provide the SAR results in a pictorial format for easy understanding to the reviewers.

Reviewer 4 Report

Comments and Suggestions for Authors

In this manuscript, Gianibbi et al. report the synthesis and antiparasitic/antibacterial properties of a series of phenothiazine-chloroquine hybrids with a diverse array of alkyl linkers and enabled by late stage click chemistry. Through systematic in vitro potency / toxicity studies as well as computer modeling of docked poses and pKa to putative biochemical targets, the authors identify a strong SAR of linker length to potency, and identify two lead compounds with potent activity. The authors propose the NDH-2 protein is the putative target of action. The paper is generally well written and the results are generally clear. The following changes, however, are necessary before the article can be suitable for publication in Antibiotics.

Abstract: Having inserted “Background”, “Methods” and “Conclusions” as in-text subtitles under Abstract is not appropriate. The authors should remove these and instead construct the abstract as a single coherent paragraph.

Lines 44-45: “No new drugs are entering the development pipeline” is an overstatement. The authors should revise this statement. There are not many, but still several notable examples of small molecules that act to counter infectious disease. For example, Gilead demonstrated earlier this decade the clinical efficacy of lenacapavir, a first in class HIV capsid inhibitor. Earlier this year, the FDA approved GSK’s gepotidacin, a first in class antibiotic for UTI’s. (Ross, et al. The Lancet 2025, 405(10489), 1608-1620). So, while it is true that there are limited examples ,it is not true that there are NO new drugs entering development or IND pipeline. 

Line 45: The limited availability of new drug candidates should reference the following review:

Bergkessel, et al. ACS Infectious Diseases 2023 9(11), 2062-2071.

Line 169: The “3” in the NaN3 (azide) should be subscripted. 

Schemes 1 & 2: These synthetic schemes are largely missing yields. It is convention for synthetic yields to be reported under each arrow.

Table 2 / Line 237: The authors only evaluate toxicity against one mammalian adherent cell line (Vero). The authors should include a human adherent normal cell line such as an MCF-10a, BEAS2B, WI-38, etc. to demonstrate whether the antiproliferative effects of these compounds are also shared in human normal cell lines. 

Line 229: The authors note that the compounds exhibit less potency against CQ-sensitive parasites, but improved potency against CQ-resistant parasites. Is this effect due to the phenathiazine motif, or due to binding of the bifunctional molecule to a different target altogether in each target organism? Is there experimental evidence that NDH2 is the correct target?

Line 425: Docking does not “prove” binding affinity - it is a computer simulation of a potential physical phenomenon. The authors should soften this language to reflect this - something like “suggest” is a more appropriate description of this result, unless the authors also have biochemical cell free binding data from laboratory experiments.

Line 712, 723: The reported isolated yield of compounds 6a and 6b is remarkably low (12-15%?). The authors should address this and either attempt to optimize the reaction to obtain higher yields, or they should provide a plausible explanation for the low yielding reaction. 

Figure S-13: This 1H NMR contains several notable impurities, including two sets of multiplets at 4 ppm. The authors must repurify this compound and report clean NMR spectra. 

Figure S-15: This spectrum has a significant impurity at 0.75 ppm (triplet?) that does not correspond to the reported structure.

General note on characterization of new compounds: It is convention for the 1H, 13C, FT-IR, and MS to be reported for each new chemical entity. However, there are no FTIR resonances reported for any of the key compounds. The authors must follow characterization conventions and add FTIR resonances to each compound’s experimental. 

Conclusions: The authors generally prepare linear, alkyl linkers conjoining the phenothiazine and chloroquine motifs. I am curious to know what might happen in potency and efficacy if a PEGylated or rigid linker were used. Certainly, in the PROTAC and RIPTAC spaces, the hydrophilicity and rigidity of the linker are enormously consequential in ultimate activity. For useful reviews, see Li, et al. J. Med. Chem. 2025, 68(3), 3420-3432; Li & Crews, Chem. Soc. Rev. 2022, 51(12), 5214-5236. The authors may find it useful to comment on this. 

Round 2

Reviewer 3 Report

Comments and Suggestions for Authors

The authors have revised and corrected the suggestions. Accept the Manuscript. 

Congratulations to all.

Author Response

We thank the reviewer for its positive feedback.

Reviewer 4 Report

Comments and Suggestions for Authors

The authors have addressed a majority of my concerns. However, Figure S23 still contains a notable solvent impurity at 3.5 ppm. The authors should address this with a new spectrum that is free from solvent impurities. 

Figure 3: A box plot is not the most appropriate to show different predicted pKa's of different compounds. I would recommend a bar graph similar to that which is indicated on Figure 4. Additionally, the two components of Figure 3 are not labeled properly. The plot on the left should be labeled A) and the right should be labeled B).

Figure 7: The amine on the linker is labeled as having no other protonatable nitrogens. However, the quinoline ring is protonatable, albeit at a different pKa. The authors should remove this comment about the linker nitrogen.

Author Response

Comment 1. Figure S23 still contains a notable solvent impurity at 3.5 ppm. The authors should address this with a new spectrum that is free from solvent impurities. 

Response 1. The Supplementary file includes the new spectrum and Materials and Methods has been modified accordingly (lines 666-670).

Comment 2. Figure 3: A box plot is not the most appropriate to show different predicted pKa's of different compounds. I would recommend a bar graph similar to that which is indicated on Figure 4. Additionally, the two components of Figure 3 are not labeled properly. The plot on the left should be labeled A) and the right should be labeled B).

Response 2. Figure 3 has been modified by adding a bar plot for pKa.

Comment 3. Figure 7: The amine on the linker is labeled as having no other protonatable nitrogens. However, the quinoline ring is protonatable, albeit at a different pKa. The authors should remove this comment about the linker nitrogen.

Response 3. Done